# Food intake enhances hippocampal sharp wave-ripples

Ekin Kaya[1]*, Evan Wegienka[1], Alexandra Akhtarzandi-Das[1], Hanh Do[1], Ada Eban-Rothschild[1], Gideon Rothschild[1,2]*

[1]Department of Psychology, University of Michigan, Ann Arbor, United States; [2]Kresge Hearing Research Institute and Department of Otolaryngology, Head and Neck Surgery, University of Michigan, Ann Arbor, United States

## eLife Assessment

This **important** study assessed the effects of food intake on sharp wave-ripples in the hippocampus of mice during subsequent sleep. **Convincing** evidence supports the conclusion that sharp wave-ripples are enhanced by food consumption. This work will likely interest researchers studying multiple functions, including memory, metabolism, and brain-body physiology.

**\*For correspondence:**
ekinkaya024@gmail.com (EK);
gid@umich.edu (GR)

**Competing interest:** The authors declare that no competing interests exist.

## Abstract
Effective regulation of energy metabolism is critical for survival. Metabolic control involves various nuclei within the hypothalamus, which receive information about the body's energy state and coordinate appropriate responses to maintain homeostasis, such as thermogenesis, pancreatic insulin secretion, and food-seeking behaviors. It has recently been found that the hippocampus, a brain region traditionally associated with memory and spatial navigation, is also involved in metabolic regulation. Specifically, hippocampal sharp wave-ripples (SWRs), which are high-frequency neural oscillations supporting memory consolidation and foraging decisions, have been shown to reduce peripheral glucose levels. However, whether SWRs are enhanced by recent feeding—when the need for glucose metabolism increases, and if so, whether feeding-dependent modulation of SWRs is communicated to other brain regions involved in metabolic regulation—remains unknown. To address these gaps, we recorded SWRs from the dorsal CA1 region of the hippocampus of mice during sleep sessions before and after consumption of meals of varying caloric values. We found that SWRs occurring during sleep are significantly enhanced following food intake, with the magnitude of enhancement being dependent on the caloric content of the meal. This pattern occurred under both food-deprived and ad libitum feeding conditions. Moreover, we demonstrate that GABAergic neurons in the lateral hypothalamus, which are known to regulate food intake, exhibit a robust SWR-triggered increase in activity. These findings identify the satiety state as a factor modulating SWRs and suggest that hippocampal-lateral hypothalamic communication is a potential mechanism by which SWRs could modulate peripheral metabolism and food intake.

## Introduction

For all organisms, from bacteria to humans, survival depends on searching for resources to maintain the internal environment in a desirable state. To obtain these resources, organisms perform stimulus-seeking behaviors wherein actions that improve bodily states are repeated while others are not. While even bacteria exhibit such coordination of behaviors based on the consequences of earlier actions (*Keegstra et al., 2022*), the evolution of the nervous system in more complex organisms has enabled the use of more sophisticated strategies to learn from the past, anticipate the future, and respond rapidly for efficient metabolic regulation.

In mammals, the hippocampus supports spatial navigation and the formation of episodic memories (*O'Keefe and Nadel, 1978*; *Scoville and Milner, 1957*), which is crucial for finding and recalling the locations of food sources. At the neurophysiological level, the hippocampus encodes the trajectory leading to food via the sequential firing of hippocampal place cells (*O'Keefe and Nadel, 1978*). During food consumption and subsequent resting periods, these same neurons replay the recent spatial trajectory in a temporally compressed manner during a highly synchronous oscillatory event called sharp wave-ripples (SWRs) (*Buzsáki, 1989*; *Diba and Buzsáki, 2007*; *Joo and Frank, 2018*; *Skaggs and McNaughton, 1996*; *Wilson and McNaughton, 1994*). SWRs communicate hippocampal activity patterns to the neocortex, supporting systems-level consolidation of memories (*Brodt et al., 2023*; *Buzsáki, 2015*; *Klinzing et al., 2019*).

In addition to the neocortex, the hypothalamus, a key regulator of homeostasis, receives dense multi-synaptic hippocampal innervation via the lateral septum (LS) (*Risold and Swanson, 1997*; *Tingley and Buzsáki, 2020*). Unlike the divergent hippocampo-cortical projections, the hippocampo-lateral septum projections are highly convergent, allowing the LS to detect only the magnitude of the SWR, but not the identity and temporal order of hippocampal neuronal spiking (*Tingley and Buzsáki, 2020*). Thus, the propagation of SWRs to hypothalamic nuclei is better suited to serve homeostatic functions rather than memory consolidation (*Buzsáki and Tingley, 2023*).

While earlier lesion and stimulation studies provided evidence that the hippocampus influences peripheral physiology and homeostatic behaviors (*Clifton et al., 1998*; *Seto et al., 1983*), direct physiological evidence demonstrating that SWRs serve metabolic functions only emerged recently. By combining in vivo electrophysiology with continuous glucose monitoring in freely behaving rats, *Tingley et al., 2021* found a negative relationship between SWR rates and subsequent peripheral glucose levels. Moreover, they showed that optogenetically induced SWRs effectively reduced peripheral glucose levels, while chemogenetic inhibition of the LS abolished the correlation between SWR rate and glucose levels. Thus, this study strongly suggests that hippocampal SWRs serve a metabolic function.

The hypothesis that hippocampal SWRs have a metabolic function raises testable predictions regarding the relationship between SWRs and food consumption. The levels of glucose, the body's primary energy source, are tightly regulated by the nervous system (*López-Gambero et al., 2019*). Specifically, around meal times, neuronal systems involved in glucose metabolism proactively and reactively coordinate the bodily response to the postprandial spike in glucose levels (*Chen et al., 2015*; *Steculorum et al., 2016*; *Woods et al., 2006*). Given that SWRs reduce peripheral glucose levels (*Tingley et al., 2021*), we set out to test the hypothesis that SWRs are enhanced following meal times as part of the systems-level response to increased glucose levels.

## Results

To determine whether food consumption modulates SWRs during subsequent sleep, we recorded the local field potential (LFP) from the dorsal CA1 region of the hippocampus, along with electroencephalographic (EEG) and electromyographic (EMG) signals, in food-restricted mice during sleep sessions before and after food intake (*Figure 1A and B*). Similar to previous studies investigating learning-dependent changes in hippocampal dynamics (*Grosmark and Buzsáki, 2016*; *Oliva et al., 2020*; *Rothschild et al., 2017*), our experimental procedure consisted of three sessions: a 2-hour sleep session ('Pre-sleep'), a subsequent food consumption session of roughly 1 hour ('Meal'), and an additional 2-hour sleep session ('Post-sleep'). Within the sleep sessions, the ongoing sleep stages were determined (wake, rapid eye movement [REM], and non-rapid eye movement [NREM], see 'Materials and methods'), and only NREM periods were included in further SWR analysis. There was no significant difference in the amount of time spent in NREM sleep in the pre and post sessions (p=0.1759, paired *t*-test, *Figure 1—figure supplement 1*). During the Meal sessions, mice received varying amounts of chow (0.3–1.5 g) in their home cage without being subjected to any instructive learning paradigm. Given that mice typically consume 4–5 g of chow per day (*Bachmanov et al., 2002*), the largest amount corresponded to about one-third of their daily food intake. In the control condition, the Meal session was replaced with a waking session of similar duration with no food delivery ('No-food'). Each mouse was tested in the Meal condition 3–12 times with varying amounts of chow and at least once in the No-food condition.

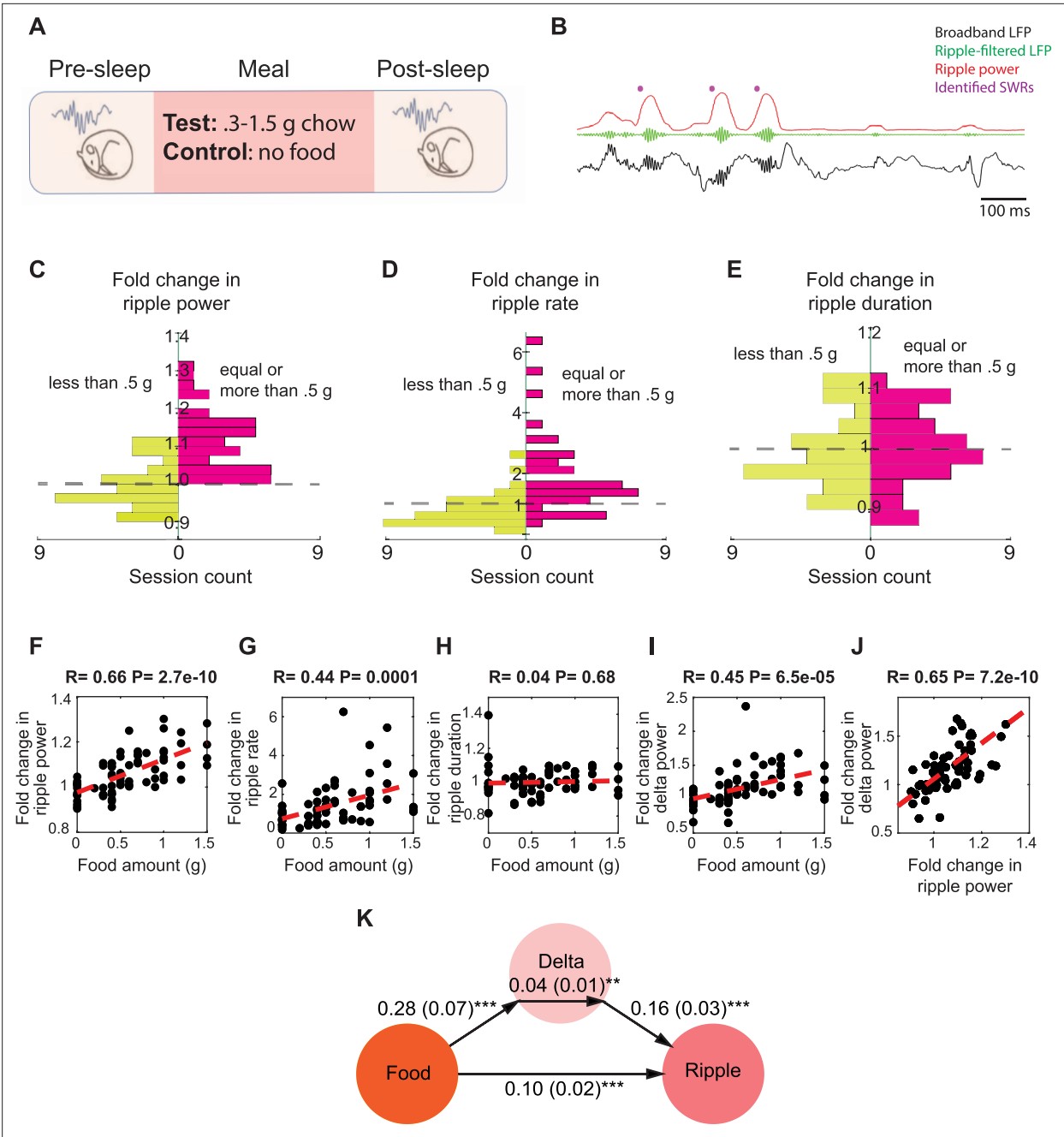

**Figure 1.** Food intake enhances sharp wave-ripples (SWRs) in sleep in a dose-dependent manner. (**A**) Experimental design. (**B**) A representative recording showing broadband hippocampal local field potential (LFP) (black), ripple-filtered LFP (150–250 Hz, green), ripple power (red), and detected SWRs (purple dots). (**C–E**) The effect of chow intake on ripple power (**C**), ripple rate (**D**), and ripple duration (**E**). Red bars represent the changes observed following delivery of equal or more than 0.5 g of chow, while the green bars represent the changes observed following delivery of less than 0.5 g. (**F–I**) Correlations between food amount and fold change in ripple power (F, $R = 0.66$, p<0.001), fold change in ripple rate (**G**, $R = 0.44$, p<0.001), fold change in ripple duration (**H**, NS) and fold change in delta power (**I**, $R = 0.45$, p<0.001). (**J**) Correlation between fold change in ripple power and fold change in delta power ($R = 0.65$. p<0.001). Panels (**F–J**) include both the experimental and control animals—the control animals appearing as having received 0 food amount. (**K**) A mediation analysis, indicating that the effect of food intake on ripple power was significant after accounting for sleep-dependent effects (correlation between the changes in delta power and ripple power: B = 0.16, SE = 0.03, p<0.001; the effect of meal size on the change in delta power: B = 0.28, SE = 0.07, p<0.001; the indirect effect of the change in delta power mediating the relationship between meal size and ripple power: B = 0.04, SE = 0.01, p<0.05, the direct effect of meal size on the change in ripple power: B = 0.10, SE = 0.02, p<0.001).

The online version of this article includes the following figure supplement(s) for figure 1:

**Figure supplement 1.** Comparison of non-rapid eye movement (NREM) sleep durations in the pre and post sessions.

To assess whether food intake modulates SWRs, we calculated the fold change in SWR statistics from pre-sleep to post-sleep in paired sessions. We found that consuming larger meals (0.5–1.5 g) significantly increased the ripple power of SWRs in all 38 session pairs, whereas smaller meals and the No-food condition did not result in a similar effect (*Figure 1C*). We observed a similar modulation in ripple rate, which was increased in 31 of 38 sessions following consumption of larger meals (*Figure 1D*), but not in ripple duration (*Figure 1E*). Smaller meals and No-food condition did not induce a consistent change in ripple rate or duration (*Figure 1D and E*). We then examined whether there is a relationship between meal size and the magnitude of change in ripple attributes. We found that the magnitude of increase in ripple power and rate, but not duration, was positively correlated with meal size (*Figure 1F–H*). These findings suggest that the power and rate of SWRs increase following feeding in a dose-dependent manner.

Since the mice in the aforementioned experiments were food restricted, we wondered whether the observed increase in ripple power after large meals might be an indirect consequence of reduced sleep quality during hunger and its restoration following feeding (*Goldstein et al., 2018*). As a measure of NREM sleep quality, we quantified cortical EEG delta power, which is also associated with SWR occurrence (*Goldstein et al., 2018*; *Mölle et al., 2006*; *Sirota et al., 2003*). While delta power indeed increased after food intake in a dose-dependent manner (*Figure 1I*), and the change in delta power was correlated with the change in ripple power (*Figure 1J*), mediation analysis revealed that the effect of meal size on ripple power remained significant even after accounting for changes in delta power (*Figure 1K*). Thus, while food intake enhanced delta power during sleep, it also led to a significant increase in ripple power beyond this effect.

To further rule out the possibility that recovery from hunger accounts for the observed increased ripple power following feeding, we conducted an additional set of experiments under ad libitum feeding conditions to ensure that sleep quality in the pre-sleep session was not affected by hunger. To encourage eating during the Meal session under ad libitum conditions, we replaced the standard chow with chocolate (0.1–1 g) (*Figure 2A*). Consistent with our previous findings, we found that chocolate consumption increased ripple power in subsequent sleep, with larger amounts of chocolate leading to greater increases in ripple power (*Figure 2B*). Moreover, the correlation between chocolate consumption and change in delta power was not significant, nor was the correlation with ripple rate or duration (*Figure 2C–E*). Further mediation analysis revealed that the change in delta power did not significantly mediate the relationship between chocolate amount and the change in ripple power (*Figure 2F*). Together, these results demonstrate that food intake enhances SWRs during sleep, even after accounting for the positive effect of food intake on sleep depth.

Next, we sought to uncover the mechanisms by which food intake modulates SWRs. Previous studies suggested that neural systems regulating food intake are responsive to both sensory and motor experiences related to feeding, such as food detection, chewing, and mechanical stretch of the gut, as well as humoral signals indicating energy availability (*Bai et al., 2019*; *Chen et al., 2015*; *Thanarajah et al., 2019*). To disentangle the factors underlying the meal enhancement of SWRs, we first tested the effects of the caloric content of meals on SWRs by comparing the effects of caloric (1 g of regular Jello) and non-caloric (1 g of sugar-free Jello, containing artificial sweeteners) foods with similar sensory qualities (*Figure 3A*). We reasoned that if their effects on SWRs differ, this would indicate a calorie dependency in the meal enhancement of SWRs, whereas no difference would suggest an experience dependency. We found that consumption of regular Jello, but not sugar-free Jello, resulted in an increase in ripple power and ripple rate compared to the No-food condition (*Figure 3B and C*). Additionally, the increase in ripple power, but not rate, was higher following consumption of regular Jello compared to sugar-free Jello (*Figure 3B and C*). No differences were found in ripple duration or cortical delta power across conditions (*Figure 3D and E*). These results suggest that the caloric content of meals, rather than the oro-/mechano-sensory experience of eating, underlies the post-meal enhancement in SWRs.

We next tested whether feeding-associated hormones contribute to the meal enhancement of SWRs (*Figure 4A*). Ghrelin, a prominent gut hormone, rises during fasting (*Toshinai et al., 2001*) and promotes increases in glucose levels and food intake (*Poher et al., 2018*). In contrast, meals trigger the release of insulin, glucagon-like peptide-1 (GLP-1), and leptin, which lower glucose levels and induce satiety (*Gutzwiller et al., 1999*; *Woods et al., 2006*). To investigate whether changes in these hormone concentrations modulate SWRs, we administered GLP-1, leptin, insulin, and ghrelin

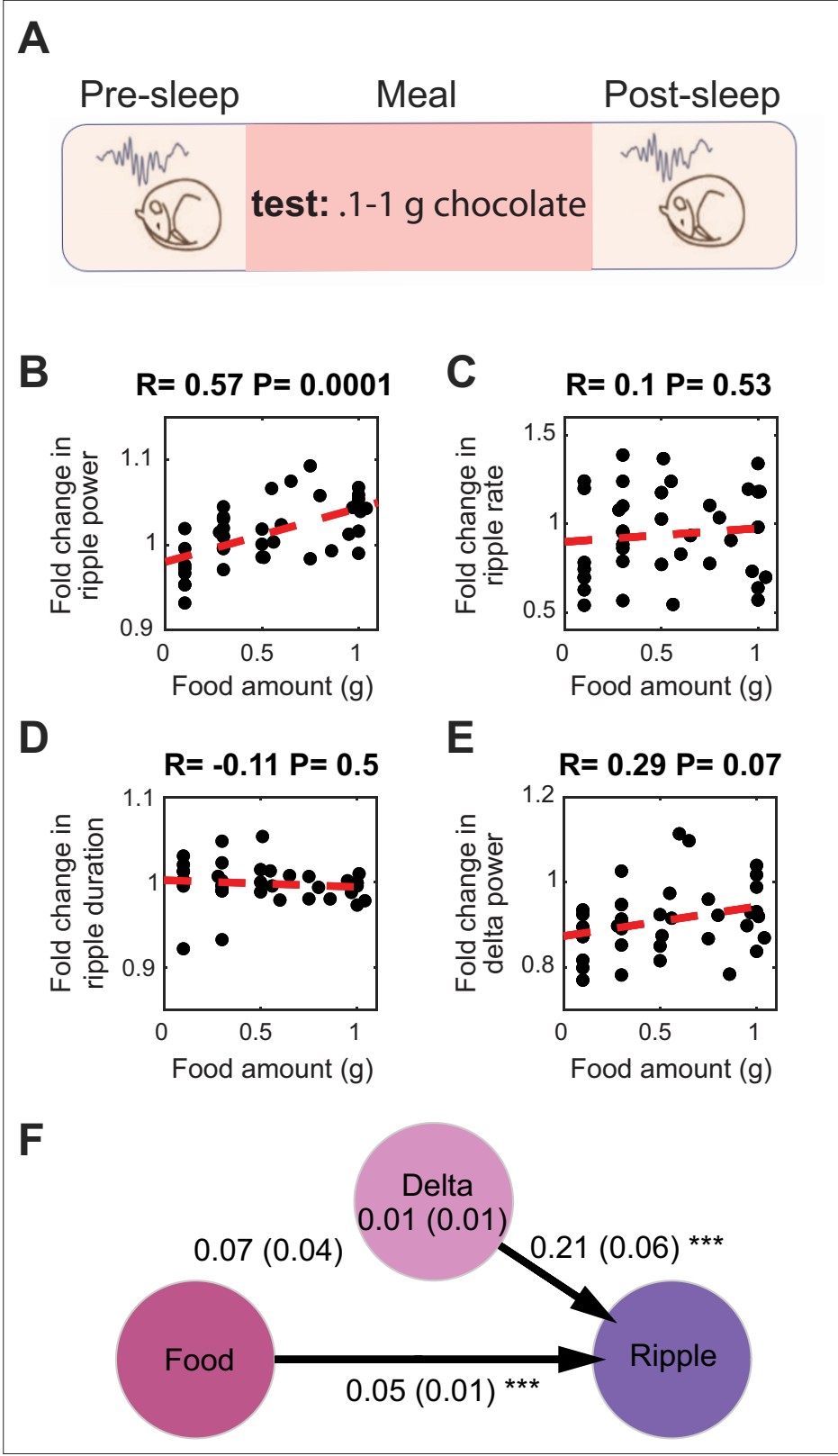

**Figure 2.** Food intake enhances sharp wave-ripples (SWRs) under ad libitum conditions. (**A**) Experimental design. (**B**) Correlation between the amount of chocolate consumed and the degree of the increase in ripple power under ad libitum conditions ($N_{animal}$ = 9, $N_{session}$ = 39, $R$ = 0.57, p=0.0001). (**C–E**) correlations between amount of chocolate consumed and fold change in ripple rate (**C**, $R$ = 0.1, p=0.53), ripple duration (**D**, $R$ = –0.11, p=0.1), and delta

*Figure 2 continued*

power (**E**, *R* = 0.29, p=0.07). (**F**) A mediation analysis showed that the effect of chocolate intake on ripple power was significant after accounting for sleep-dependent effects (the indirect effect of the change in delta power mediating the relationship between meal size and ripple power: B = 0.01, SE = 0.01, p>0.05, the direct effect of meal size on the change in ripple power: B = 0.05, SE = 0.01, p<0.001).

to mice at different doses under ad libitum and food-restricted conditions. Administration of GLP-1, leptin, and insulin did not produce robust, consistent, and dose-dependent effects on SWR attributes (*Figure 4B–G*). On the other hand, ghrelin administration resulted in a dose-dependent decrease in ripple rate under ad libitum conditions (*Figure 4H*) and a decrease in ripple power under food-restricted conditions (*Figure 4I*). While ghrelin administration also reduced cortical delta power, mediation analysis revealed that the effect of ghrelin on ripple power remained significant even after accounting for change in cortical delta power under food-restricted conditions (*Figure 4Ii and ii*). These results suggest that ghrelin has a dose-dependent negative effect on ripple rate and power, which exceeds its influence on sleep quality. In contrast, we did not find consistent modulation of SWRs following GLP-1, insulin, or leptin administration, despite their known effects on hippocampal memories and synaptic transmission (*Abbas et al., 2009*; *McNay et al., 2010*; *Oomura et al., 2006*). It is possible that specific aspects of our experimental design (e.g., focusing on the 2-hour post-sleep period, administering only one hormone at a time, or using subcutaneous delivery) may have occluded potential effects of these hormones on SWRs.

Lastly, as our findings reveal that SWRs are modulated by food-related signals, we tested the hypothesis that SWRs can, in turn, influence brain regions involved in feeding regulation. Among the hypothalamic nuclei that participate in feeding regulation (*Anand and Brobeck, 1951*; *Stuber and Wise, 2016*), the lateral hypothalamus (LH) receives the densest di-synaptic input from the dorsal hippocampus via the lateral septum (*Risold and Swanson, 1997*; *Tingley and Buzsáki, 2020*). To test whether SWRs affect neuronal activity in the LH, we combined LFP recordings from the hippocampus with optical recordings from GABAergic neurons in the LH using fiber photometry. We used Vgat-IRES-Cre mice (n = 5) and transduced LH neurons with the pGP-AAV5-syn-FLEX-jGCaMP8f-WPRE vector (*Figure 5A and B*). We found that LH GABAergic neurons exhibit a robust and consistent

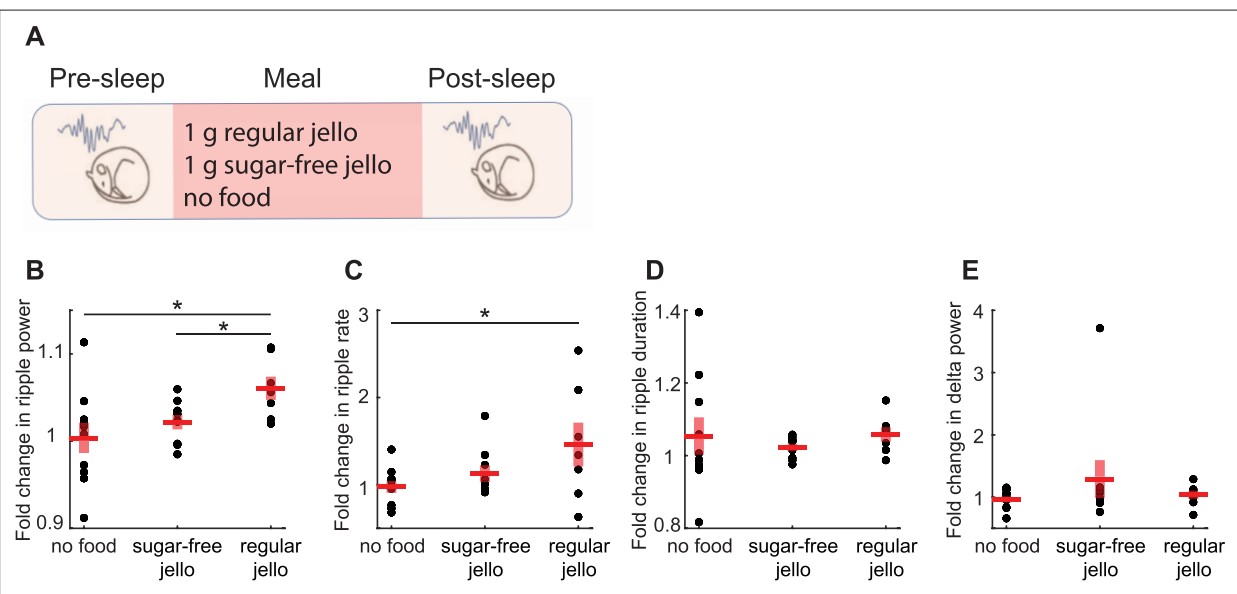

**Figure 3.** Caloric content of meals, rather than the experience of food intake, drives the enhancement of sharp wave-ripples (SWRs). (**A**) Experimental design (N_regular = 7, N_sugar-free = 9, N_nofood = 10). (**B**) Consumption of regular Jello resulted in a greater increase in ripple power compared to non-caloric Jello (t(14) = 2.59, p=0.02) and the control condition (t(15) = 2.39, p=0.03), with no significant difference between sugar-free Jello and the control condition (p=0.38). (**C**) Regular Jello consumption increased SWR rate compared to the control condition (t(15) = 2.18, p=0.046), with no other significant differences across conditions (ps>0.05). No differences were detected in the SWR duration or delta power across conditions (**D, E** ps>0.05). Red lines indicate mean, and shaded red areas indicate standard error. Datapoints within each panel represent different animals.

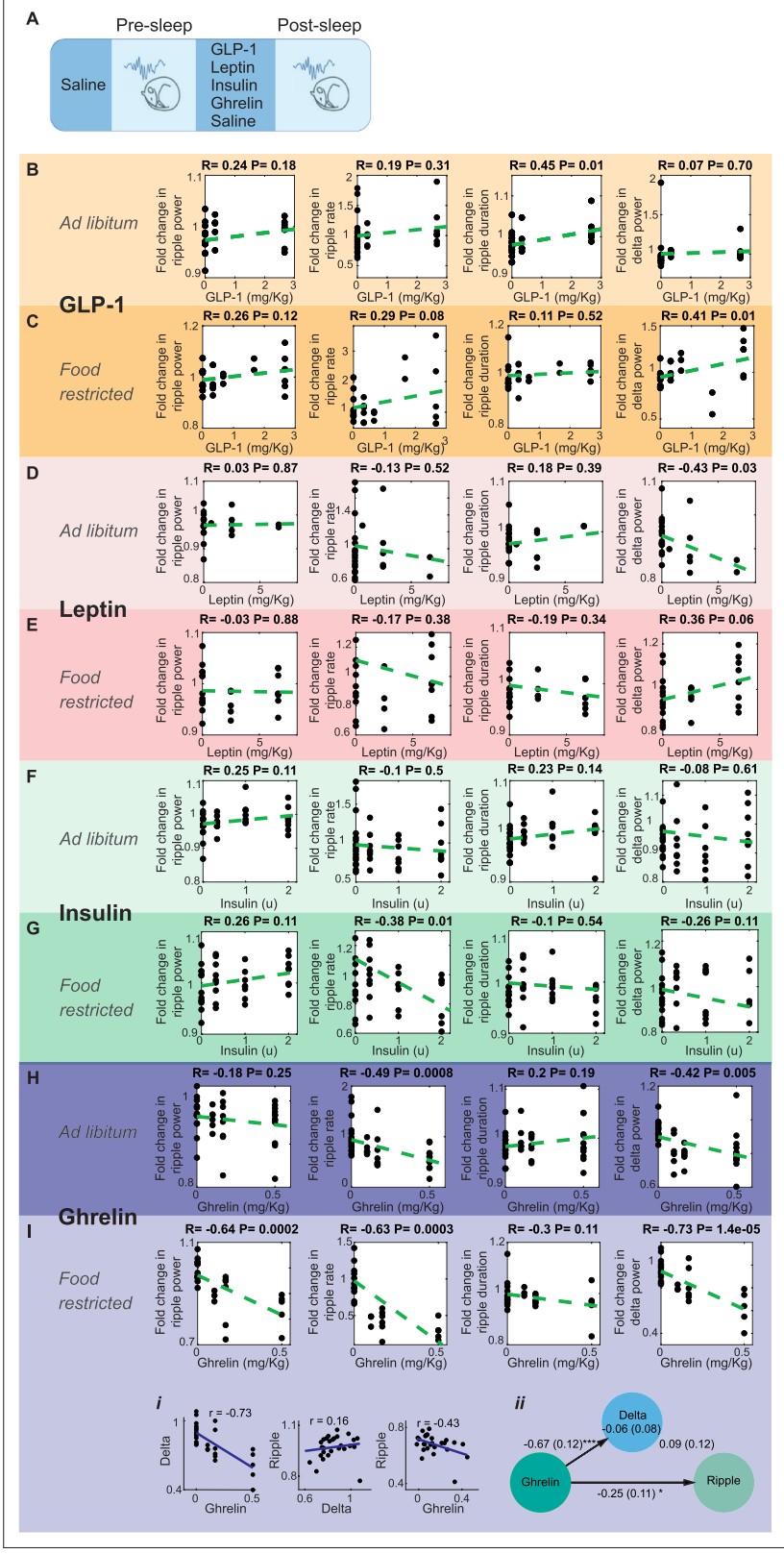

**Figure 4.** Effect of systemic administration of hormones on sharp wave-ripple (SWR) properties. (**A**) Experimental design. No dose-dependent effects were detected in ripple power, rate, or duration following the administration of GLP-1 (**B, C**), leptin (**D, E**), or insulin (**F**) (ps>0.05), except a decrease in ripple rate following insulin administration under food-restricted conditions (**G**, *R* = –0.38, p=0.01). Ghrelin administration caused a dose-

*Figure 4 continued on next page*

*Figure 4 continued*

dependent decreased in SWR rate ($R = -0.49$, p=0.0008) and delta power ($R = -0.42$, p=0.005) under ad libitum conditions (**H**), and decreased ripple power ($R = -0.64$, p=0.0002), ripple rate ($R = -0.63$, p=0.0003), and delta power ($R = -0.73$, p=0.00014) under food-restricted conditions (**I**). Mediation analysis revealed that ghrelin's effect on ripple power remained significant after accounting for changes in cortical delta power under food-restricted conditions (**Ii, Iii**, direct effect of ghrelin on ripple power: B = -0.25, SE = 0.11, p<0.05).

---

increase in activity, peaking approximately 320 ms after SWR onset (*Figure 5C–J*, *Figure 5—figure supplement 1*). These findings strongly suggest that SWRs, which we found to be modulated by feeding, are also associated with a consistent and robust increase in population activity in the LH, a central brain region for metabolic regulation.

## Discussion

Regulation of energy intake and expenditure is vital for animals. While food intake provides essential fuel for the organism, it can also elevate plasma glucose and other nutrient levels above optimal ranges for physiological functions. This elevation triggers coordinated neural and peripheral responses that lower glucose levels by promoting glucose uptake in peripheral tissues and inhibiting further food intake (*Chen et al., 2015*; *López-Gambero et al., 2019*; *Steculorum et al., 2016*; *Woods et al., 2006*). Here, we show that hippocampal SWRs, previously linked to reductions in peripheral glucose levels (*Tingley et al., 2021*), are enhanced after food intake. Specifically, we compared SWRs occurring during sleep before and after meals and found a calorie-dependent increase in the ripple power and SWR rate. This enhancement exceeded the increase in sleep quality and was evident under both ad libitum and food-restricted conditions. Our caloric vs. non-caloric Jello experiments further revealed that the enhancement in SWRs was driven by caloric intake rather than the experience of eating. Moreover, we found that GABAergic neurons in the LH, a region implicated in the regulation of both food intake and peripheral metabolism (*Anand and Brobeck, 1951*; *López-Gambero et al., 2019*; *Shimazu, 1987*), exhibit a robust and consistent SWR-triggered increase in activity.

We conducted our experiments during the light phase and focused our analyses on sleep because awake SWRs primarily occur during *offline* states (i.e., resting) (*Buzsáki, 2015*), which are reduced under food restriction due to increasing physical activity (*Chowdhury et al., 2015*). However, a bidirectional relationship between hunger and sleep also exists (*Danguir and Nicolaidis, 1979*; *Rechtschaffen and Bergmann, 1995*), where hunger increases arousal and impairs sleep quality (*Danguir and Nicolaidis, 1979*; *Goldstein et al., 2018*). In line with this, we found that cortical delta power was enhanced following meals in previously food-restricted mice. Nonetheless, mediation analyses showed that food intake still had a significant effect on ripple power, even after accounting for the restoration of sleep quality. To further exclude the possibility that our results were a by-product of impaired sleep caused by prior food restriction, we performed additional experiments under ad libitum food access. Although we did not observe a change in ripple rate in these experiments, we found a dose-dependent increase in ripple power following chocolate consumption, replicating our earlier findings under food-restricted conditions. Notably, the extent of SWR modulation was weaker under ad libitum conditions, similar to the asymmetric hypothalamic response to food in sated versus fasted animals (*Chen et al., 2015*).

Conversely, a poor night's sleep leads to increased appetite and impaired metabolic regulation the subsequent day (*Hogenkamp et al., 2013*; *Klingenberg et al., 2013*; *Schmid et al., 2015*). Chronic sleep disturbances, on the other hand, elevate the risk of long-term conditions such as diabetes and metabolic syndrome (*Knutson et al., 2007*). Although the neural mechanisms underlying these processes remain largely unknown, a recent study linked stronger coupling of sleep spindles and cortical slow-wave activity to improved glucose metabolism regulation the subsequent day in humans (*Vallat et al., 2023*). Given that about 75% of SWRs occur during NREM sleep in rats (*Tingley et al., 2021*), and that SWRs are often triggered when slow waves and spindles are coupled (*Sirota et al., 2003*; *Staresina et al., 2015*), this suggests that SWRs can play a role in the connection between sleep and next-day metabolic regulation.

The hippocampus, which integrates environmental and interoceptive signals along with past experiences to support spatial navigation and memory, has also been shown to influence food intake

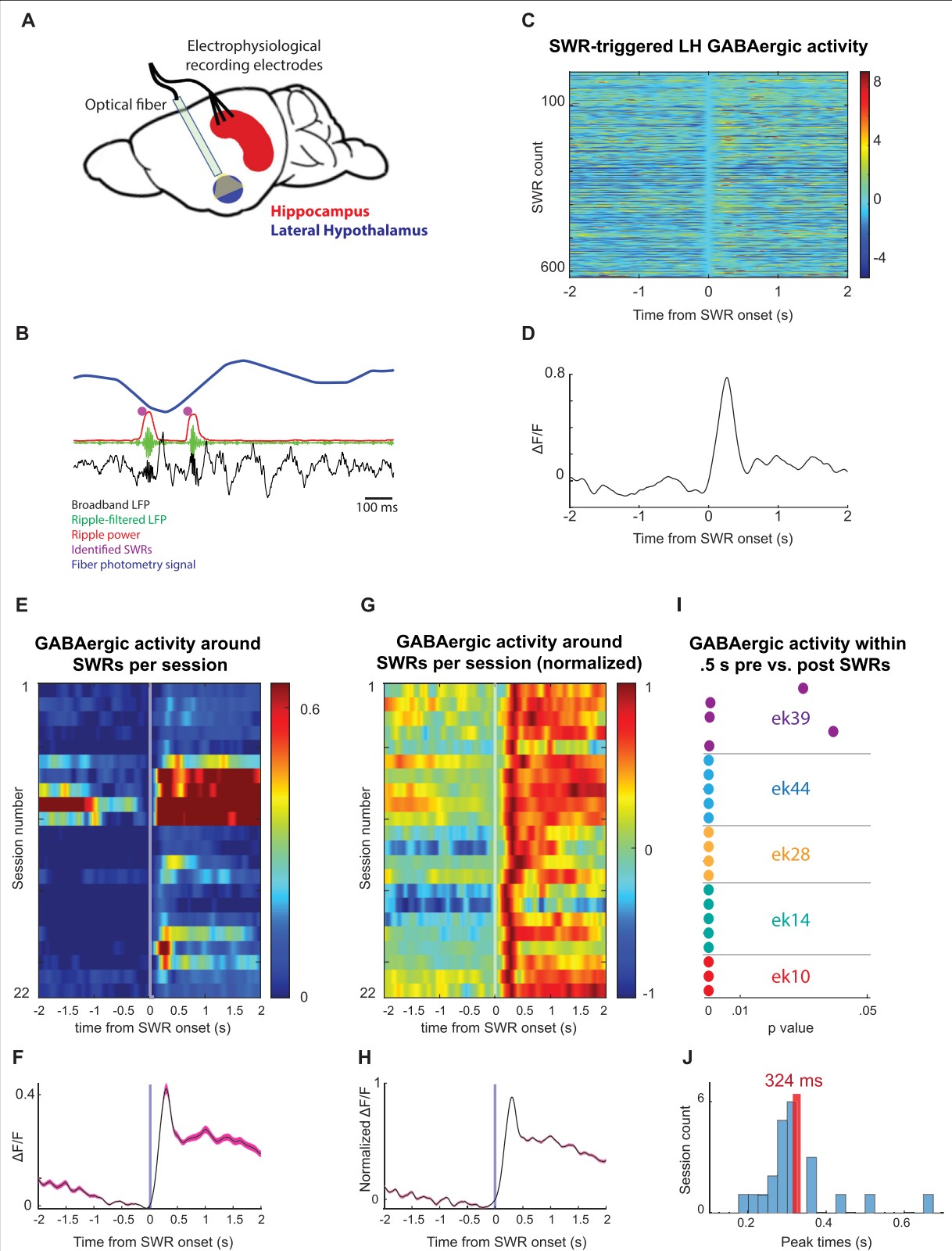

**Figure 5.** Lateral hypothalamic GABAergic neuronal populations exhibit an increase in activity following sharp wave-ripples (SWRs). (**A**) Experimental design. (**B**) A representative recording showing broadband hippocampal local field potential (LFP) (black), ripple-filtered LFP (150–250 Hz, green), ripple power (red), detected SWRs (purple dots), and the fiber photometry signal from lateral hypothalamus (LH) GABAergic neurons (FP, blue). (**C**) Example traces of LH GABAergic neuronal activity around SWRs. (**D**) Mean trace of peri-SWR LH GABAergic activity from one session. (**E**) Mean SWR-triggered

*Figure 5 continued on next page*

*Figure 5 continued*

LH GABAergic activity profiles per session. (**F**) The mean of the SWR-triggered LH GABAergic activity profiles across sessions. (**G, H**) Similar to (**E, F**) but for normalized traces. (**I**) Statistical significance of LH GABAergic activity within 0.5 s before and after SWRs per session. (**J**) Peak time of LH GABAergic activity relative to SWR onset. Shaded error bars (pink) indicate standard deviation.

The online version of this article includes the following figure supplement(s) for figure 5:

**Figure supplement 1.** Additional quantification of sharp wave-ripple (SWR)-triggered increase in lateral hypothalamus (LH) GABAergic activity.

(*Kanoski and Grill, 2017*; *Stevenson and Francis, 2017*). Notably, HM, a famous patient who lost the ability to form new memories after the removal of his hippocampi, also exhibited irregular eating habits and difficulties with interoceptive judgments (*Hebben et al., 1985*). Earlier rodent studies similarly showed that hippocampal lesions increase meal frequency in rats (*Clifton et al., 1998*). More recently, optogenetic inhibition of the hippocampus following meals was shown to decrease the latency to the next meal and increase its size (*Hannapel et al., 2019*). These findings suggest that hippocampal-dependent memories of recent meals may suppress appetite.

Another study identified dopamine-2 receptor neurons in the dorsal CA3 region of the hippocampus as being responsive to meals and food cues, with inhibition and activation of these neurons increasing and decreasing food intake, respectively (*Azevedo et al., 2019*). Additional research has highlighted the role of the hippocampus in utilizing peripheral signals related to energy balance for meal regulation (*Davidson and Jarrard, 1993*; *Kanoski and Grill, 2017*; *Wee et al., 2024*). For example, the administration of leptin into both the ventral and dorsal hippocampus (*Kanoski et al., 2011*), and exendin-4 (a GLP-1 agonist) into the ventral hippocampus (*Hsu et al., 2015*), have been shown to reduce feeding. Conversely, ghrelin signaling in the ventral hippocampus has been shown to increase meal size through a descending LH-hindbrain circuit (*Suarez et al., 2020*). A recent study found distinct neural populations in the dorsal hippocampus that show increased activity following the intake of either fats or sugars, and these populations were involved in memory for the location of the respective foods, as well as in food intake (*Yang et al., 2025*).

While previous studies employing pharmacological and viral approaches have revealed the role of the hippocampus in the regulation of food intake, understanding at the physiological level has remained limited. Peripheral hormones that rise during the postprandial period, such as insulin, amylin and GLP-1, not only reduce plasma glucose levels but also act as satiety signals that inhibit food intake (*Woods et al., 2006*). Given this dual inhibitory effect of postprandial signals on both glucose levels and food intake, we propose that SWRs could also play an inhibitory role in regulating food intake. Future studies could explore this possibility by using closed-loop suppression of SWRs (*Girardeau et al., 2009*). This approach could also be used to test whether SWRs underlie the link between sleep loss and subsequent impairment in glucose metabolism and increase in appetite (*Hogenkamp et al., 2013*; *Schmid et al., 2015*; *Klingenberg et al., 2013*). While we did not find consistent modulation of SWRs following GLP-1, insulin, or leptin administration, we believe this is likely due to specific aspects of our experimental design, such as focusing on the 2-hour post-sleep period, administering only one hormone at a time, or the use of subcutaneous delivery of the hormones.

While our focus was on metabolic factors affecting SWRs, our results may also have implications for the interaction between memory and metabolism. It is widely recognized that memory tasks guided by food reward enhance SWRs (*Ambrose et al., 2016*; *Singer and Frank, 2009*). This modulation, thought to support the consolidation of memory for the correct trajectory leading to food, could also be influenced by food intake and may modulate feeding-related processes. Our Jello experiments showed that the enhancement in SWRs following food intake is dependent on caloric content of meals rather than the experience of eating. However, one might argue that only salient experiences (i.e., consumption of food with calories, but not without) are consolidated, and thus the enhancement of SWRs following regular Jello consumption could reflect the memory consolidation of a salient experience. Previous studies have shown that animals can learn the nutrition content of different foods (*Yamamoto and Ueji, 2011*). Since our mice had been habituated to all foods before the experiments (regular Jello, sugar-free Jello, and chocolate), they might have anticipated that sugar-free Jello would not provide calories (i.e., a less salient experience) or raise glucose levels. However, the enhancement of SWRs for memory consolidation typically requires novel events. As our mice were well-habituated to both the experimental paradigm and the Jello, it is unlikely that the enhancement in SWRs reflects a memory consolidation process. One potential experiment to bypass mnemonic processes could

involve direct gastric infusion of food. Further studies could also investigate whether food consumption after specific events enhances consolidation through the modulation of SWRs.

We found that the SWR property that was most strongly modulated by food is ripple power, followed by SWR rates, while SWR durations were unmodulated by feeding. This is interesting in light of prior studies showing that memory consolidation is linked to an increased rate (*Eschenko et al., 2008*) and duration (*Fernández-Ruiz et al., 2019*) of SWRs or replay events (though reactivation strength also increases; *Giri et al., 2019*). Ripple power is strongly correlated with the degree of synchronized recruitment of hippocampal populations (*Csicsvari et al., 1999*; *Schomburg et al., 2012*). In contrast, increased SWR rates following learning allow repeated reactivation of recent experience (*Eschenko et al., 2008*; *Giri et al., 2019*), while long-duration SWRs have been proposed to facilitate replay of particularly long spatial trajectories (*Fernández-Ruiz et al., 2019*). In alignment with the previous finding that replay content is not communicated to subcortical regions involved in metabolic regulation (*Tingley and Buzsáki, 2020*), ripple power may be the key attribute relevant for homeostatic signaling, while replay content, manifesting in increased SWR rate and duration, underlies the mnemonic function of SWRs.

We propose that the temporal coordination between the onset of SWRs and the increase in LH population activity may represent a mechanism by which SWRs influence peripheral metabolism and food intake. GABAergic lateral septal neurons can facilitate the relay of SWRs to the LH by providing windows of disinhibition (*Risold and Swanson, 1997*; *Tingley et al., 2021*; *Wong et al., 2016*). While our fiber-photometry recordings from the LH provided a population-level activity signal, future studies could utilize single-cell resolution imaging methods to uncover network-level organization and dynamics (*Aharoni et al., 2019*; *Lee and Rothschild, 2021*; *Vivaldo et al., 2023*; *Vogt, 2022*). Furthermore, while our investigation of hippocampal-LH communication was limited to the timescale of seconds around SWRs, whether and how this communication changes across the timescale of hours following feeding remains to be determined. Finally, the causal relationship between SWRs and LH activity, as well as the functional significance of these interactions, remain to be investigated.

In summary, our study demonstrates that SWRs are sensitive to the body's hunger state and are modulated by food intake. Whether SWRs can, in turn, influence food intake and the specific mechanisms underlying this potential modulation remain to be elucidated.

## Materials and methods

### Animals

All experimental procedures followed the US National Institutes of Health Guide for the Care and Use of Laboratory Animals and were approved by the University of Michigan's Institutional Animal Care and Use Committee. We used C57BL/6 wild-type mice (n = 8, seven male and one female) and Vgat-IRES-Cre (n = 9 male mice; The Jackson laboratory, #016962) heterozygous mice bred in-house. The mice were housed under a 12:12 light-dark cycle.

### Surgeries

Mice (7–16 weeks old) were anesthetized with a ketamine-xylazine mixture (100 and 10 mg/kg, respectively; administered via intraperitoneal injection [IP]) and received lidocaine and carprofen (4 mg/kg and 5 mg/kg, respectively) prior to being placed in a stereotaxic frame (David Kopf Instruments, Tujunga, CA). During the surgery, the mice were kept under isoflurane anesthesia (~1% in $O_2$).

For hippocampal electrophysiology recordings, an array of four tungsten wires (50 micron, California Fine Wire Company) were implanted in the CA1 region of the dorsal hippocampus (AP: –2 mm, ML: 1.5 mm, DV: –1.4 mm). Additionally, an EMG wire was inserted into the neck muscle and an EEG and a ground screw were placed above the parietal cortex and the cerebellum, respectively.

Fiber photometry procedures were done as described previously (*Eban-Rothschild et al., 2020*; *Sotelo et al., 2024*). Briefly, 400 nL of pGP-AAV-syn-FLEX-jGCaMP8f-WPRE (Addgene) (2.6e$^{12}$ vg/mL) were injected into the LH (AP: –1.2 mm, ML: 1.15 mm, DV: –5 mm) and mono fiber optic probes (400 μm diameter, 0.48 NA, Doric Lenses, Inc, Quebec, Canada) were implanted just above the LH (AP: –1.5 mm, ML: 1.15 mm, DV: –4.6 mm).

Mice were given several days to recover from surgery. They were then habituated to handing and individually housed in custom Plexiglas recording cages (39.4 cm × 28.6 cm × 19.3 cm) with open

tops to ensure that the LFP cable could reach the mice and allow them to be freely behaving without restricted movements. We recorded mice until 1 year of age.

## Data processing, analyses, and sleep scoring

Electrophysiological and optical recordings were conducted using the Tucker Davis Technology system, sampled at 3 kHz and 1 kHz respectively. Data was visualized with Synapse or SynapseLite software. Sleep scoring was done manually using the AccuSleep toolbox (https://github.com/zeke-barger/AccuSleep; *Barger and Frye, 2025*) using the EEG and EMG signals. NREM periods were detected based on high EEG delta power and low EMG power, REM periods were detected based on high EEG theta power and low EMG power, and wake periods were detected based on high EMG power. Only NREM periods were included for subsequent SWR detection, quantification, and analyses (in particular, reported SWR rates reflect the number of SWRs per second of NREM sleep). Sessions with less than 30 minutes of sleep were excluded from further analyses.

To detect SWRs, we filtered the raw signal between 150 and 250 Hz and took an envelope of the squared filtered signal. We identified SWRs by identifying the events that reached at least 5 standard deviations above the mean and that were above 1 standard deviation for at least 40 ms. In case two events occurred within 40 ms or less, we discarded the second event. Ripple power threshold calculations were done together for pre-sleep and post-sleep. We calculated the delta power by taking the envelope of the squared filtered signal between 0.5 and 4 Hz on the EEG channel during NREM sleep.

For pre-sleep and post-sleep comparisons, we calculated average ripple power, ripple rate, ripple duration, and delta power for each pre and post session. Change from pre-sleep to post-sleep was calculated as change = mean (variable$_{postsleep}$)/mean(variable$_{presleep}$). Mediation analyses were done using the Mediation toolbox (https://github.com/canlab/MediationToolbox; *Ashar et al., 2025*).

For SWR-LH interaction analyses, we interpolated the fiber photometry signal to match its length to the length of the LFP. For each SWR, we extracted the surrounding 4 second windows of the fiber photometry signal and centered the fiber photometry signal at the ripple onset to zero. We detected the SWR-triggered peaks in fiber photometry signal within the 1 second following SWRs. For statistical comparisons, we calculated the integrals of the 0.5-second windows before and after each SWR and performed a paired *t*-test on these integrals. For the summary plots, we normalized each session by its own maximum value.

## Food restriction and experimental procedures

All experiments were conducted during the light phase. Chow experiments were conducted under food-restricted conditions in which animals did not have access to food for 20–24 hours preceding the experiments. Experimental steps were as follows: mice were left to sleep for 2 hours (pre-sleep). Then they were woken up and given a piece of regular chow (Meal). The amount of chow varied between 0.3 and 1.5 g. To calculate the exact amount of consumption, we removed and weighed any remaining chow pieces before the start of the post-sleep session which lasted for another 2 hours. Each gram of chow contained 4.1 kcal. For chocolate experiments, mice went through the same experimental procedure except that they had ad libitum food access throughout the experiment and received varying amounts (0.1–1 g) of Reese's Peanut Butter Cups instead of chow. Each gram of chocolate contained 5 kcal.

Jello experiments were performed under food-restricted conditions by using regular or sugar-free strawberry flavored Jello. Each 10 g of Jell-O gelatin mix was dissolved in 13 mL water for regular Jello meals, and each 10 g of sugar-free strawberry flavored Jell-o gelatin mix was dissolved in 111 mL water for sugar-free meals. The mixtures were refrigerated (4°C) until firm. 1 g of regular and sugar-free Jello meals contained 1.7 and 0 kcals, respectively. The sugar-free Jello contained artificial sweeteners (aspartame and acesulfame potassium). The Jello meals were given on plastic weighing boats and weighed before and after the trial to calculate the exact amount of consumption.

## Hormone experiments

The experiments were conducted under 24-hour food restriction or ad libitum food access conditions. No food pieces were present in the cages during the recordings. Before pre-sleep sessions, mice received a subcutaneous injection of 0.1 mL saline. Before post-sleep sessions, they were administered with either 0.1 mL saline or various doses of hormones, including ghrelin (0.1, 0.167, or 0.5 mg/

kg body weight, GenScript), leptin (0.3, 3, 6.67 mg/kg, Peprotech), insulin (0.5, 1, or 2 u/kg body weight, Sigma-Aldrich, I5500), or GLP-1 (0.33, 0.67, 1.67, or 2.67 mg/kg body weight, Tocris, Cat# 2082), all dissolved in 0.1 mL saline.

## Acknowledgements

This work was supported by a Boehringer Ingelheim Fonds fellowship (EK), a National Institute of Health grant R01NS131821 (AE-R and GR), a National Institute of Health grant R01NS129874 (GR and AE-R) and a University of Michigan Alzheimer's Disease Research Center grant P30AG072931 (GR). The authors thank Alika Sulaman for her help with fiber photometry experiments and Zeinab Mansi and Lindsay Cain for their assistance with histology.

## Additional information

### Funding

| Funder | Grant reference number | Author |
|---|---|---|
| Boehringer Ingelheim Fonds | Fellowship | Ekin Kaya |
| National Institute of Neurological Disorders and Stroke | R01NS131821 | Ada Eban-Rothschild Gideon Rothschild |
| National Institute of Neurological Disorders and Stroke | R01NS129874 | Ada Eban-Rothschild Gideon Rothschild |
| University of Michigan | P30AG072931 | Gideon Rothschild |

The funders had no role in study design, data collection and interpretation, or the decision to submit the work for publication.

### Author contributions

Ekin Kaya, Conceptualization, Data curation, Formal analysis, Funding acquisition, Investigation, Writing – original draft; Evan Wegienka, Alexandra Akhtarzandi-Das, Hanh Do, Data curation; Ada Eban-Rothschild, Conceptualization, Supervision, Funding acquisition, Project administration, Writing – review and editing; Gideon Rothschild, Conceptualization, Resources, Formal analysis, Supervision, Funding acquisition, Investigation, Project administration, Writing – review and editing

### Author ORCIDs

Ekin Kaya ⬤ https://orcid.org/0000-0001-9235-4954
Ada Eban-Rothschild ⬤ https://orcid.org/0000-0001-5816-1315
Gideon Rothschild ⬤ https://orcid.org/0000-0002-9700-1020

### Ethics

This study was performed in strict accordance with the recommendations in the Guide for the Care and Use of Laboratory Animals of the National Institutes of Health. All of the animals were handled according to the University of Michigan Institutional Animal Care and Use Committee, specifically following animal protocol PRO00011694.

Reviewer #1 (Public review): https://doi.org/10.7554/eLife.105059.3.sa1
Reviewer #2 (Public review): https://doi.org/10.7554/eLife.105059.3.sa2
Reviewer #3 (Public review): https://doi.org/10.7554/eLife.105059.3.sa3
Author response https://doi.org/10.7554/eLife.105059.3.sa4

## Additional files

### Supplementary files
MDAR checklist

### Data availability
Data and code used in this study are available in figshare.

The following dataset was generated:

| Author(s) | Year | Dataset title | Dataset URL | Database and Identifier |
|---|---|---|---|---|
| Rothschild G | 2025 | Data and code for Kaya et al. (2025) | https://doi.org/10.6084/m9.figshare.28691093 | figshare, 10.6084/m9.figshare.28691093 |

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
