## [Editor Report · eLife Assessment]

This **important** study assessed the effects of food intake on sharp wave-ripples in the hippocampus of mice during subsequent sleep. **Convincing** evidence supports the conclusion that sharp wave-ripples are enhanced by food consumption. This work will likely interest researchers studying multiple functions, including memory, metabolism, and brain-body physiology.

---

## [Referee Report · Reviewer #1 (Public review)]

Summary:

This manuscript by Kaya et al. studies the effect of food consumption on hippocampal sharp wave ripples (SWRs) in mice. The authors use multiple foods and forms of food delivery to show that the frequency and power of SWRs increase following food intake, and that this effect depends on the caloric content of food. The authors also studied the effects of administration of various food-intake-related hormones on SWRs during sleep, demonstrating that ghrelin negatively affects SWR rate and power, but not GLP-1, insulin, or leptin. Finally, the authors use fiber photometry to show that GABAergic neurons in the lateral hypothalamus, increase activity during a SWR event.

Strengths:

The experiments in this study seem to be well performed, and the data are well presented, visually. The data support the main conclusions of the manuscript that food intake enhances hippocampal SWRs. Taken together, this study is likely to be impactful to the study of the impact of feeding on sleep behavior, as well as the phenomena of hippocampal SWRs in metabolism.

Weaknesses:

None

---

## [Referee Report · Reviewer #2 (Public review)]

Summary:

Kaya et al uncover an intriguing relationship between hippocampal sharp wave-ripple production and peripheral hormone exposure, food intake, and lateral hypothalamic function. These findings significantly expand our understanding of hippocampal function beyond mnemonic processes and point a direction for promising future research.

Strengths:

Some of the relationships observed in this paper are highly significant. In particular, the inverse relationship between GLP1/Leptin and Insulin/Ghrelin are particularly compelling as this aligns well with opposing hormone functions on satiety.

---

## [Referee Report · Reviewer #3 (Public review)]

Summary:

The manuscript by Kaya et al. explores the effects of feeding on sharp wave-ripples (SWRs) in the hippocampus, which could reveal a better understanding of how metabolism is regulated by neural processes. Expanding on prior work that showed that SWRs trigger a decrease in peripheral glucose levels, the authors further tested the relationship between SWRs and meal consumption by recording LFPs from the dorsal CA1 region of the hippocampus before and after meal consumption. They found an increase in SWR magnitude during sleep after food intake, in both food-restricted and ad libitum fed conditions. Using fiber photometry to detect GABAergic neuron activity in the lateral hypothalamus, they found increased activity locked to the onset of SWRs. They conclude that the animal's satiety state modulates the amplitude and rate of SWRs, and that SWRs modulate downstream circuits involved in regulating feeding.

The authors have addressed prior requests for revision and clarification, and provide a convincing case for SWRs being modulated by satiety state. These experiments provide an important step forward in understanding how metabolism is regulated in the brain. The study will likely be of great interest in the field of learning and memory while carrying broader implications for understanding brain-body physiology.

---

## [Author Response]

The following is the authors’ response to the original reviews

**Public Reviews:**

**Reviewer #1 (Public review):**
Summary:This manuscript by Kaya et al. studies the effect of food consumption on hippocampal sharp wave ripples (SWRs) in mice. The authors use multiple foods and forms of food delivery to show that the frequency and power of SWRs increase following food intake, and that this effect depends on the caloric content of food. The authors also studied the effects of the administration of various food-intake-related hormones on SWRs during sleep, demonstrating that ghrelin negatively affects SWR rate and power, but not GLP1, insulin, or leptin. Finally, the authors use fiber photometry to show that GABAergic neurons in the lateral hypothalamus, increase activity during a SWR event.Strengths:The experiments in this study seem to be well performed, and the data are well presented, visually. The data support the main conclusions of the manuscript that food intake enhances hippocampal SWRs. Taken together, this study is likely to be impactful to the study of the impact of feeding on sleep behavior, as well as the phenomena of hippocampal SWRs in metabolism.Weaknesses:Details of experiments are missing in the text and figure legends. Additionally, the writing of the manuscript could be improved.

We thank the reviewer for their favorable assessment of the work and its potential impact. We have added all requested details in the text and figure legends and revised the wording of the manuscript to improve its clarity.

**Reviewer #2 (Public review):**
Summary:Kaya et al uncover an intriguing relationship between hippocampal sharp wave-ripple production and peripheral hormone exposure, food intake, and lateral hypothalamic function. These findings significantly expand our understanding of hippocampal function beyond mnemonic processes and point a direction for promising future research.Strengths:Some of the relationships observed in this paper are highly significant. In particular, the inverse relationship between GLP1/Leptin and Insulin/Ghrelin are particularly compelling as this aligns well with opposing hormone functions on satiety.Weaknesses:I would be curious if there were any measurable behavioral differences that occur with different hormone manipulations.

We thank the reviewer for their favorable assessment of the work and its contribution to our understanding of non-mnemonic hippocampal function. Whether there are behavioral differences that occur following administration of the different hormones is a great question, yet unfortunately our study design did not include fine behavioral monitoring to the degree that would allow answering it. While some previous studies have partially addressed the behavioral consequences of the delivery of these hormones (and we reference these studies in our Discussion), how these changes may interact with the hippocampal and hypothalamic effects we observe is a very interesting next step.

**Reviewer #3 (Public review):**
Summary:The manuscript by Kaya et al. explores the effects of feeding on sharp wave-ripples (SWRs) in the hippocampus, which could reveal a better understanding of how metabolism is regulated by neural processes. Expanding on prior work that showed that SWRs trigger a decrease in peripheral glucose levels, the authors further tested the relationship between SWRs and meal consumption by recording LFPs from the dorsal CA1 region of the hippocampus before and after meal consumption. They found an increase in SWR magnitude during sleep after food intake, in both food restricted and ad libitum fed conditions. Using fiber photometry to detect GABAergic neuron activity in the lateral hypothalamus, they found increased activity locked to the onset of SWRs. They conclude that the animal's satiety state modulates the amplitude and rate of SWRs, and that SWRs modulate downstream circuits involved in regulating feeding. These experiments provide an important step forward in understanding how metabolism is regulated in the brain. However, currently, the paper lacks sufficient analyses to control for factors related to sleep quality and duration; adding these analyses would further support the claim that food intake itself, as opposed to sleep quality, is primarily responsible for changes in SWR activity. Adding this, along with some minor clarifications and edits, would lead to a compelling case for SWRs being modulated by a satiety state. The study will likely be of great interest in the field of learning and memory while carrying broader implications for understanding brain-body physiology.Strengths:The paper makes an innovative foray into the emerging field of brain-body research, asking how sharp wave-ripples are affected by metabolism and hunger. The authors use a variety of advanced techniques including LFP recordings and fiber photometry to answer this question. Additionally, they perform comprehensive and logical follow-up experiments to the initial food-restricted paradigm to account for deeper sleep following meal times and the difference between consumption of calories versus the experience of eating. These experiments lay the groundwork for future studies in this field, as the authors pose several follow-up questions regarding the role of metabolic hormones and downstream brain regions.

We thank the reviewer for their appreciation and constructive review of the work.

Weaknesses:Major comments:(1) The authors conclude that food intake regulates SWR power during sleep beyond the effect of food intake on sleep quality. Specifically, they made an attempt to control for the confounding effect of delta power on SWRs through a mediation analysis. However, a similar analysis is not presented for SWR rate. Moreover, this does not seem to be a sufficient control. One alternative way to address this confound would be to subsample the sleep data from the ad lib and food restricted conditions (or high calorie and low calorie, etc), to match the delta power in each condition. When periods of similar mean delta power (i.e. similar sleep quality) are matched between datasets, the authors can then determine if a significant effect on SWR amplitude and rate remains in the subsampled data.

This is an important point that we believe we addressed in a few complementary ways. First, the mediation analysis we implemented measures the magnitude and significance of the contribution of food on SWR power after accounting for the effects of delta power, showing a highly significant food-SWR contribution. While the objective of subsampling is similar, mediation is a more statistically robust approach as it models the relationship between food, SWR power, and delta power in a way that explicitly accounts for the interdependence of these variables. Further, subsampling introduces the risk of losing statistical power by reducing the sample size, due to exclusion of data that might contain relevant and valuable information. Mediation analysis, on the other hand, uses the full dataset and retains statistical power while modeling the relationships between variables more holistically. However, as we were not satisfied with a purely analytical approach to test this issue, we carried out a new set of experiments in ad-libitum fed mice, where there is no concern of food restriction impairing sleep quality in the presleep session. In these conditions food amount also significantly correlated with, and showed significant mediation of, the SWR power change. Finally, we acknowledge and discuss this point in the Discussion, highlighting that given the known relationship between cortical delta and SWRs, it is challenging to fully disentangle these signals.

(2) Relatedly, are the animals spending the same amount of time sleeping in the ad lib vs. food restricted conditions? The amount of time spent sleeping could affect the probability of entering certain stages of sleep and thus affect SWR properties. A recent paper (Giri et al., Nature, 2024) demonstrated that sleep deprivation can alter the magnitude and frequency of SWRs. Could the authors quantify sleep quantity and control for the amount of time spent sleeping by subsampling the data, similar to the suggestion above?

Following the reviewer’s comment, we have quantified and compared the amount of time spent in NREM sleep in the Pre and Post session pairs in which the animals were food restricted, with 0-1.5 g of chow given between the sleep sessions. We found that there was no significant difference in the amount of time spent in NREM sleep in the Pre and Post sessions. We have added this result to the Results section of the manuscript and as a new Supplementary Fig. 1.

Additionally, we have added details to the Methods section that were missing in the original submission that are relevant to this point. Specifically, within the sleep sessions, the ongoing sleep states were scored using the AccuSleep toolbox (https://github.com/zekebarger/AccuSleep) using the EEG and EMG signals. NREM periods were detected based on high EEG delta power and low EMG power, REM periods were detected based on high EEG theta power and low EMG power, and Wake periods were detected based on high EMG power. Importantly, only NREM periods were included for subsequent SWR detection, quantification and analyses (in particular, reported SWR rates reflect the number of SWRs per second of NREM sleep).

(3) Plot 5I only reports significance but does not clearly show the underlying quantification of LH GABAergic activity. Upon reading the methods for how this analysis was conducted, it would be informative to see a plot of the pre-SWR and post-SWR integral values used for the paired t-test whose p-values are currently shown. For example, these values could be displayed as individual points overlaid on a pair of boxand-whisker plots of the pre- and post-distribution within the session (perhaps for one example session per mouse with the p-value reported, to supplement a plot of the distribution of p-values across sessions and mice). If these data are non-normal, the authors should also use a non-parametric statistical test.

We have generated the summary plots the reviewer requested and have now included them in Supplementary Fig. 2.

Minor comments:(4) A brief explanation (perhaps in the discussion) of what each change in SWR property (magnitude, rate, duration) could indicate in the context of the hypothesis may be helpful in bridging the fields of metabolism and memory. For example, by describing the hypothesized mechanistic consequence of each change, could the authors speculate on why ripple rate may not increase in all the instances where ripple power increases after feeding? Why do the authors speculate that ripple duration does not increase, given that prior work (Fernandez-Ruiz et al. 2019) has shown that prolonged ripples support enhanced memory?

This is an interesting point and we have added a section to the Discussion to discuss it (pg. 17, last paragraph)

(5) The authors suggest that "SWRs could modulate peripheral metabolism" as a future implication of their work. However, the lack of clear effects from GLP-1, leptin and insulin complicates this interpretation. It might be informative for readers if the authors expanded their discussion of what specific role they speculate that SWRs could play in regulating metabolism, given these negative results.

We have added a section to the Discussion proposing potential reasons for this point (pg. 16, last paragraph)

**Recommendations for the authors:**

**Reviewer #1 (Recommendations for the authors):**
Major Comments:(1) The experiments involve very precise windows of time for sleeping and eating that seem impossible to control. For example, the authors state that for the experiments in Figure 1, there was a 2-h sleep period, followed by a 1-h feeding period, followed by another 2-h sleep period. Without sleep deprivation procedures or other environmental manipulations, how can these periods be so well-defined? Even during the inactive period, mice typically don't sleep for 2-h bouts at once, and the addition of food would not likely lead to an exact 1-h period of wakefulness in the middle. The validity of these experimental times would be more believable if the authors provided much more data on these sessions. For example, the authors could provide a table or visual display of data for the actual timing of the pre-sleep, eating, and post-sleep phases with exact time measurements and/or visual display of sleep versus wakefulness.

This is an important point, which we were not clear enough about in the original submission. While the durations of the Pre-sleep, Wake and Post-sleep sessions were indeed 2 h, 1 h and 2 h respectively, the animals did not actually sleep during the entirety of the sleep sessions. Importantly, we performed sleep state scoring on all sessions, and only analyzed identified NREM sleep for all SWR analyses. Following the reviewer’s comment (and that of Reviewer 1), we have quantified and compared the amount of time spent in NREM sleep in the Pre and Post session pairs in which the animals were food restricted and 0-1.5 g of chow were given between the sleep sessions. We found that there was no significant difference in the amount of time spent in NREM sleep in the Pre and Post sessions. We have added this result to the Results section of the manuscript and as a new Supplementary Fig. 1.

Additionally, we have added details to the Methods section that were missing in the original submission that are relevant to this point. Specifically, within the sleep sessions, the ongoing sleep states were scored using the AccuSleep toolbox (https://github.com/zekebarger/AccuSleep) using the EEG and EMG signals. NREM periods were detected based on high EEG delta power and low EMG power, REM periods were detected based on high EEG theta power and low EMG power, and Wake periods were detected based on high EMG power. Importantly, only NREM periods were included for subsequent SWR detection, quantification and analyses (in particular, reported SWR rates reflect the number of SWRs per second of NREM sleep).

(2) I may have missed this (although I tried searching in the text and figure legend), but the authors did not state the difference between green versus red bar colors in Figure 1 C-E. For Figures 1 F-J, do the individual dots represent both the test (fed) animals and control animals, or just the test animals?

We thank the reviewer for the opportunity to clarify these points. Red bars in Fig. 1C-E represent the SWR changes observed following delivery of equal or more than 0.5 g of chow, while the green bars represent the changes observed following delivery of less than 0.5 g. Fig. 1F-J includes both the experimental and control animals- the control animals appearing as having received 0 food amount. This information has now been added to the figure legend.

(3) For the jello experiments in Figure 3, was there only 1 trial per animal? Previous studies show that animals learn the caloric value of jello after subsequent trials, so whether or not multiple trials took place in each animal is important for interpretation of the results.

In Figure 3, the datapoints within each panel represent different animals and this information has now been added to the figure legend. Nevertheless, the animals were previously habituated to all foods, including regular jello, sugar-free jello and chocolate. While we consider it unlikely that this prior experience was sufficient to underlie the differential effects on SWRs, we cannot fully rule out the possibility that it provided some ability to predict the caloric value and consequences of the different foods. We have added details to the acknowledgement of this point in the Discussion (pg. 17, second paragraph).

(4) The experiments in Figure 5 are informative but don't relate to the experiments in the rest of the study. It is difficult to interpret their meaning given that these experiments take place over seconds while the other experiments take place over hours. Some attempt should be made to bridge these experiments over the timescales relevant for the behaviors studied in Figures 1-4.

We have now further acknowledged and discussed the point that our investigation is limited to the timescale of seconds around SWRs, and thus identified a potential communication channel, but whether and how this communication changes across hours following feeding remains for future studies (pg. 18, second paragraph).

(5) Figure 5B should depict the x-axis in seconds, not an arbitrary set of times from a recording.

We have replaced these with a time scale bar.

Minor Comments:(6) The writing of the manuscript can be improved in many places:Sometimes the writing could be more precise. For example, the Abstract states: "hippocampal sharp wave ripples (SWRs)... have been shown to influence peripheral glucose metabolism." Could this be written in a more informative way, rather than just staying "has been shown to influence?" A few more words would provide a lot more information. Similarly, at the end of the Introduction: "we set out to test the hypothesis that SWRs are modulated following meal times as part of the systems-level response to changing metabolic needs." This is not a strong hypothesis... could it be written to boldly state how the SWRs will be modulated (increase or decrease) and provide more assertive information?The writing can be grandiose at times. Phrases such as "life is a continuous journey" or "the hypothalamus is a master regulator of homeostasis" are a bit sophomoric and too colloquial.Finally, a representative recording should be referred to as just that-a "representative recording," as opposed to a "snippet," which is also colloquial. This word is used in the figure legends to Figures 1 and 5, and misspelled as "sinpper" in Figure 1

We have reworded all these sentences and phrases to make them clearer, more concrete and more formal.

(7) The methods state that the study used both male and female mice. Were they used in equal numbers across experiments?

Only one female was used in the final dataset, and we have corrected the wording accordingly.

**Reviewer #2 (Recommendations for the authors):**
Great paper!

Thanks!

**Reviewer #3 (Recommendations for the authors):**
Below are some minor requests for clarification, including in figures:(1) Fig. 5H y-axis should say "normalized dF/F."

Done

(2) Fig. 1B is missing a y-axis label. It may be clearer to display separate y-axis scale bars for each component (SWR envelope, ripple-filtered amplitude, etc).

Done

(3) Please include labels for brain areas and methodological components in Fig. 5A.

Done

(4) Should Fig. 5B have the same y-axis or scale bars as 1B?

We have edited the figure labels and legends to be visually similar

(5) In Fig. 5J, is the y-axis a count of sessions?

Yes, we have added that to the y-axis label

(6) Could the authors please clarify whether the sugar-free jello was sweetened with an artificial sweetener? If so, this is a robust control for the rewarding nature of the two jellos, so a quick clarification would highlight this strength of the experiment.

We thank the reviewer for this great point. Indeed, the sugar free jello contained artificial sweeteners (Aspartame and Acesulfame Potassium). We have added this information to the Results and Methods.

(7) It appears in Fig. 5 that there may be a reliable dip in activity **at** the time of SWR onset, followed by the increase afterward, as shown in the example FP trace and the individual ripple-triggered traces. Is this indeed the case, and does this dip fall significantly below baseline? This characterization would be interesting, but I acknowledge is not necessarily crucial to the study to include.

This would indeed be an interesting finding, but upon examination and statistical testing, we found that this is not the case. We believe this may appear as such due to the normalization of the traces.

(8) The authors mention a reduction in ripple rate following insulin under food restriction as the only significant effect for insulin, GLP-1, and leptin, yet there was also a significant increase (at p<0.05) in ripple duration for GLP-1 in the ab lib condition. Is this not considered noteworthy?

This is a fair point and we have reworded the description of this result to simply state that there were no robust, consistent, dose-dependent effects of GLP-1, leptin and insulin on SWR attributes.